# Topological antichiral surface states in a magnetic Weyl photonic crystal

Xiang Xi [1,6], Bei Yan [1,6], Linyun Yang [1,6], Yan Meng [1], Zhen-Xiao Zhu [1], Jing-Ming Chen [1], Ziyao Wang[1], Peiheng Zhou [2], Perry Ping Shum [1], Yihao Yang [3], Hongsheng Chen [3], Subhaskar Mandal[4], Gui-Geng Liu [4] ✉, Baile Zhang [4,5] ✉ & Zhen Gao [1] ✉

Chiral edge states that propagate oppositely at two parallel strip edges are a hallmark feature of Chern insulators which were first proposed in the celebrated two-dimensional (2D) Haldane model. Subsequently, counterintuitive antichiral edge states that propagate in the same direction at two parallel strip edges were discovered in a 2D modified Haldane model. Recently, chiral surface states, the 2D extension of one-dimensional (1D) chiral edge states, have also been observed in a photonic analogue of a 3D Haldane model. However, despite many recent advances in antichiral edge states and chiral surface states, antichiral surface states, the 2D extension of 1D antichiral edge states, have never been realized in any physical system. Here, we report the experimental observation of antichiral surface states by constructing a 3D modified Haldane model in a magnetic Weyl photonic crystal with two pairs of frequency-shifted Weyl points (WPs). The 3D magnetic Weyl photonic crystal consists of gyromagnetic cylinders with opposite magnetization in different triangular sublattices of a 3D honeycomb lattice. Using microwave field-mapping measurements, unique properties of antichiral surface states have been observed directly, including the antichiral robust propagation, tilted surface dispersion, a single open Fermi arc connecting two projected WPs and a single Fermi loop winding around the surface Brillouin zone (BZ). These results extend the scope of antichiral topological states and enrich the family of magnetic Weyl semimetals.

Since its introduction in the 1980s, the landmarked 2D Haldane model[1] on a honeycomb lattice with time-reversal-symmetry breaking has been a paradigmatic model featuring distinct topological phases of matter. It has inspired the discovery of Chern insulators characterized by chiral edge states[2–5] that propagate nonreciprocally and oppositely along two parallel strip edges, and the quantum spin Hall effect that can be interpreted as two overlaid Haldane models related by time-reversal symmetry[6,7]. The Haldane model and its robust chiral edge

[1]Department of Electronic and Electrical Engineering, Southern University of Science and Technology, 518055 Shenzhen, China. [2]National Engineering Research Center of Electromagnetic Radiation Control Materials, Key Laboratory of Multi-spectral Absorbing Materials and Structures of Ministry of Education, University of Electronic Science and Technology of China, 611731 Chengdu, China. [3]Interdisciplinary Center for Quantum Information, State Key Laboratory of Modern Optical Instrumentation, ZJU-Hangzhou Global Science and Technology Innovation Center, College of Information Science and Electronic Engineering, ZJU-UIUC Institute, Zhejiang University, 310027 Hangzhou, China. [4]Division of Physics and Applied Physics, School of Physical and Mathematical Sciences, Nanyang Technological University, Singapore 637371, Singapore. [5]Centre for Disruptive Photonic Technologies, The Photonics Institute, Nanyang Technological University, Singapore 639798, Singapore. [6]These authors contributed equally: Xiang Xi, Bei Yan, Linyun Yang. ✉e-mail: guigeng001@e.ntu.edu.sg; blzhang@ntu.edu.sg; gaoz@sustech.edu.cn

states have not only revolutionized condensed matter physics[1-8], but also sparked the emerging field of topological photonics[9-15] and topological acoustics[16-20]. Recently, by subtly reversing the next-nearest-neighbor (NNN) couplings in one triangular sublattice, the original 2D Haldane model is modified and the two Dirac points at $K$ and $K'$ shift frequencies, giving rise to tiled edge dispersions connecting two frequency-shifted Dirac points and counterintuitive antichiral edge states that propagate in the same direction at opposite strip edges with gapless bulk states supplying the required counter-propagating modes[21-25]. Significantly, the discovery of antichiral edge states in the 2D modified Haldane model arises broad research interests and deeply enriches the realm of topological physics.

More recently, a photonic analog of a 3D Haldane model[26] was reported that behaves as 3D photonic Chern insulators or magnetic Weyl semimetals, depending on the time-reversal symmetry and inversion symmetry breaking strengths. Both 3D photonic Chern insulators[26-28] and magnetic Weyl semimetals[29-34] support robust chiral surface states that propagate nonreciprocally and oppositely along two parallel cuboid surfaces. Inspired by recent advances in antichiral edge states and chiral surface states, it is highly desirable and scientifically meaningful to realize topological antichiral surface states that propagate in the same direction along two parallel cuboid surfaces. However, antichiral surface states have never been reported (even in theory) in any physical system.

In this work, we report the experimental observation of antichiral surface states in a 3D magnetic Weyl photonic crystal, which incorporates gyromagnetic materials placed in external magnetic fields to break the time-reversal symmetry, based on a 3D modified Haldane model. More challengingly, we must adopt precise on-site modulation of magnetization[23,25] to implement opposite magnetic fluxes on different sublattice sites $A$ and $B$ in a 3D honeycomb lattice, in stark contrast to the original 2D/3D Haldane model with the same magnetic fluxes on all sublattice sites[11,26]. The oppositely imposed magnetic fluxes on different triangular sublattices will shift the frequencies of two pairs of WPs and induce antichiral surface states with tiled Fermi arc surface dispersions that connect two frequency-shifted WPs, similar to the antichiral edge states in the 2D modified Haldane model. Using microwave near-field imaging measurements, we observe a unique photonic magnetic Weyl semimetal with two pairs of frequency-shifted WPs. Consequently, on two opposite and parallel surfaces of the magnetic Weyl photonic crystal, the Fermi arc surface states that connect two pairs of frequency-shifted WPs acquire titled dispersions and propagate nonreciprocally in the same direction, in contrast to the previously reported chiral surface states that propagate oppositely along two parallel cuboid surfaces[26,28]. More interestingly, we also demonstrate that this magnetic Weyl semimetal can host an evolution of surface state iso-frequency contour from a single open Fermi arc (the simplest Fermi arc configuration) connecting two projected WPs to a single surface Fermi loop[35] winding around the surface BZ, and finally to a single Fermi arc connecting the other two projected WPs. To our knowledge, the topological antichiral surface states associated with two pairs of frequency-shifted WPs in a magnetic Weyl semimetal have never previously been explored.

## Results

### Design of a 3D magnetic Weyl photonic crystal
We start with a 3D gyromagnetic photonic crystal featuring a gapless band structure with two pairs of frequency-shifted type-II WPs. The conceptual illustration of the 3D gyromagnetic photonic crystal with topological antichiral surface states (red and blue arrows) propagating nonreciprocally along the same direction on opposite surfaces is shown in Fig. 1a, with its honeycomb unit cell presented in Fig. 1b. The unit cell consists of a perforated metallic plate (yellow color) and two gyromagnetic rods (red and blue cylinders) sandwiched by two pairs of permanent magnets (silver cylinders) to implement precise on-site

magnetization modulation. By setting the configurations of permanent magnets, the gyromagnetic rods in two different sublattices of $A$ and $B$ sites are magnetized along $-z$ and $+z$ directions (marked as blue and red arrows in Fig. 1b), respectively. The permanent magnets are coated with high-conductivity materials and thus can be treated as perfect electric conductors (PEC) in simulations (see "Methods" for detail). The layers of gyromagnetic rods and permanent magnets are stacked periodically in the vertical direction and separated by perforated metallic plates to induce interlayer couplings and ensure the fundamental eigenmodes are $E_z$-polarized. As shown in Fig. 1c, the metallic plates are perforated with inequivalent patterns around two corners of the hexagonal unit cell, which will introduce unequal interlayer couplings for sublattices $A$ and $B$ and break the in-plane inversion symmetry. The proposed 3D gyromagnetic photonic crystal can be well qualitatively described by a 3D modified Haldane model (see "Methods" and Supplementary Information for details).

Figure 1d shows the simulated bulk band structure of the 3D gyromagnetic photonic crystal, which hosts two pairs of ideal type-II WPs[36-38] (marked by purple and orange spheres) because there exist no other trivial bands at the WPs frequency 8.22 GHz at momenta $k = (4\pi/3a, 0, \pm 0.38\pi/h)$ along the KH line, and 7.85 GHz at momenta $k = (-4\pi/3a, 0, \pm 0.38\pi/h)$ along the K'H' line (see Supplementary Information for the details of modeling in the simulations). And the topological transition (band inversion) of the bulk band structure occurs at these WPs, which can be characterized by the Berry curvature and valley Chern number calculated in the $k_x$-$k_y$ plane with the varying of $k_z$ (see Supplementary Information for details). This band structure is different from all previously reported non-magnetic[29,30] or magnetic[26,31-34] Weyl semimetals, since the two pairs of WPs exhibit a significant frequency shift. Note that this novel phenomenon can only occur in a time-reversal-symmetry broken system under oppositely imposed magnetic fluxes on different sublattices $A$ and $B$. Moreover, for each pair of WPs along the KH or K'H' line, they carry opposite topological charges (Chern numbers) +1 and −1 (see the calculation of WPs charge in "Methods"). Due to the opposite magnetization directions and unequal interlayer couplings on sublattice sites $A$ and $B$, the two opposite and parallel surfaces of the 3D gyromagnetic photonic crystal are distinct and denoted as A-type and B-type surfaces, respectively, as illustrated in Fig. 1a. Figure 1f shows the calculated Fermi arc surface states dispersions of A-type (blue color) and B-type (red color) surfaces, respectively. Note that the surface states dispersions of A-type and B-type surfaces in the $xz$ plane depend significantly on the cutting position along the $y$ direction (Two different boundary selections are analyzed in detail in Supplementary Information). Due to the frequency shift of two WPs pairs, both Fermi arc surface state dispersions tilt along $+k_x$ direction with positive group velocities, indicating that two surface states on A-type and B-type surfaces propagate along the same direction and thus dubbed as antichiral surface states. To further identify the tilted antichiral surface state dispersions, we plot the frequency-dependent surface dispersion curves along $k_x$ direction with $k_z = 0\pi/h$, $k_z = 0.38\pi/h$, and $k_z = 1\pi/h$, respectively, as shown in Fig. 1g–i. Specifically, at $k_z = 0\pi/h$ ($k_z = 1\pi/h$), for the A-type surface, the bulk bandgap opens and the surface states (blue solid line) connect the projections of the lower (upper) bulk bands. By contrast, the situations are reversed for the B-type surface (red solid line). More interestingly, at $k_z = 0.38\pi/h$, the projected bulk states of the first and second bands form two degenerate points at $k_x = 0.67\pi/a$ (at 7.85 GHz) and $k_x = 1.33\pi/a$ (at 8.22 GHz) which are exactly the surface projections of WPs. And the Fermi arc surface state dispersions of A-type (blue dashed line) and B-type (red dashed line) surfaces are degenerate and obliquely connect the two projected frequency-shifted WPs (W$_{1+}$ and W$_{2-}$), similar to that of the antichiral edge states connecting two projected frequency-shifted Dirac points in the 2D modified Haldane model[21-25]. The reason for the degeneracy of two antichiral surface states at fixed $k_z = 0.38\pi/h$ is because of the mirror symmetry in the 2D

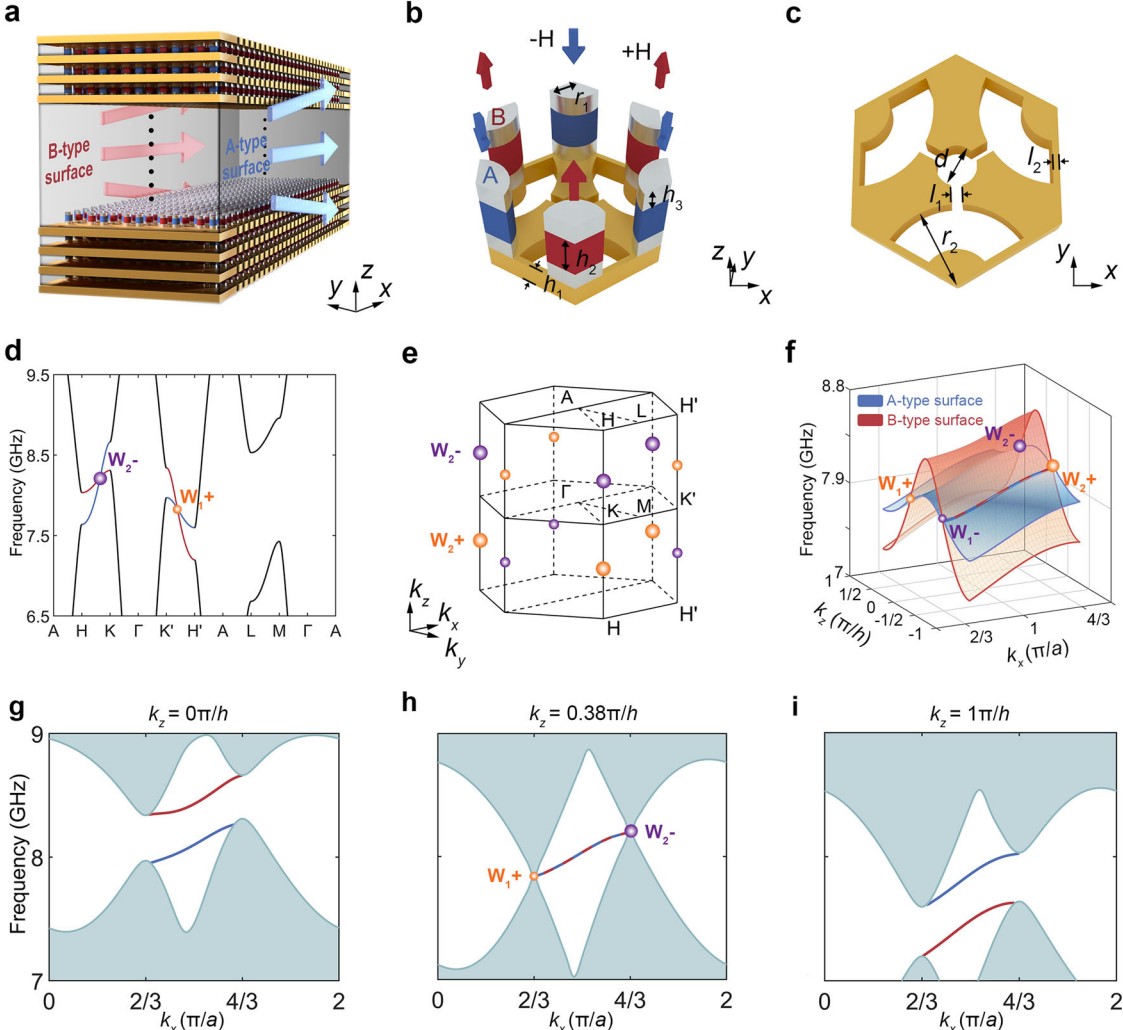

**Fig. 1 | Design of a 3D magnetic Weyl photonic crystal. a** Conceptual illustration of antichiral surface states (red and blue arrows) that propagate in the same direction on opposite surfaces of a 3D gyromagnetic Weyl photonic crystal. **b** The unit cell of 3D gyromagnetic photonic crystal consisting of two sandwiched gyromagnetic rods (red and blue fan-shaped rods) sitting on the perforated metallic plate. The gyromagnetic rods at different sublattice sites of the honeycomb lattice are oppositely magnetized by pairs of permanent magnets (silver fan-shaped rods) with reversed biasing direction. The red (blue) arrows indicate the biased magnetic fluxes along $+z$ ($-z$) directions. The lattice constants in the $xy$-plane and $z$-direction are $a = 7.5$ mm and $h = 6.5$ mm, and the other geometrical parameters are $r_1 = 1.3$ mm, $h_1 = 2.5$ mm, $h_2 = 2$ mm and $h_3 = 1$ mm, respectively. **c** Top view of a unit cell of perforated metallic plates with geometrical parameters $r_2 = 2.9$ mm, $d = 1.5$ mm, $l_1 = 0.4$ mm, $l_2 = 0.25$ mm. **d** Calculated bulk band structure of the magnetic Weyl photonic crystals along high-symmetry lines. Orange and purple spheres represent two pairs of frequency-shifted WPs carrying opposite topological charges of $+1$ and $-1$. **e** 3D bulk Brillouin zone with ideal type-II WPs. **f** Simulated 2D surface dispersions of A-type (blue curved sheet) and B-type (red curved sheet) surfaces. **g**–**i** Simulated surface dispersions for fixed values of **g** $k_z = 0\pi/h$, **h** $k_z = 0.38\pi/h$ and **i** $k_z = 1\pi/h$, respectively. The blue (red) lines indicate the A-type (B-type) surface state dispersion, respectively, and cyan regions represent the projected bulk states.

modified Haldane model (see detailed analysis in Supplementary Information).

## Experimental demonstration of a 3D magnetic Weyl photonic crystal

Now we start experimentally demonstrating the frequency-shifted WPs pairs in a 3D magnetic Weyl photonic crystal. The fabricated experimental sample is shown in Fig. 2a, which consists of 30 layers of perforated copper plates and dielectric foams inserted with gyromagnetic rods sandwiched between permanent magnets. Figure 2b, c show the partially enlarged photo of the perforated copper plates and dielectric foams inserted with gyromagnetic rods and magnets, respectively. The opposite magnetizations at different sublattice sites are distinguished by red and blue colors. The gyromagnetic rods are sandwiched between a pair of magnets with corresponding magnetization

direction and then are inserted into the dielectric foam to fix their positions, as shown in Fig. 2d, e, respectively. To measure the bulk band structure of the 3D gyromagnetic photonic crystal, as schematically shown in the left upper inset of Fig. 2a, a dipole source antenna is inserted into the center of the sample, and a dipole probe antenna that is fixed to a robotic arm is inserted into the air holes one by one to map the complex electric field distributions ($E_z$ component) in the middle $xz$ plane. After applying Fourier transform to the measured complex field distributions from real space to reciprocal space, we obtain the measured projected bulk band structure in the projected surface BZ (see lower right inset of Fig. 2a), as shown in Fig. 2f, which is consistent with the calculated projected bulk band structure (white solid lines). From both the measured and simulated results we can observe two pairs of WPs (purple and orange spheres) with a frequency shift. To our knowledge, this type of magnetic Weyl

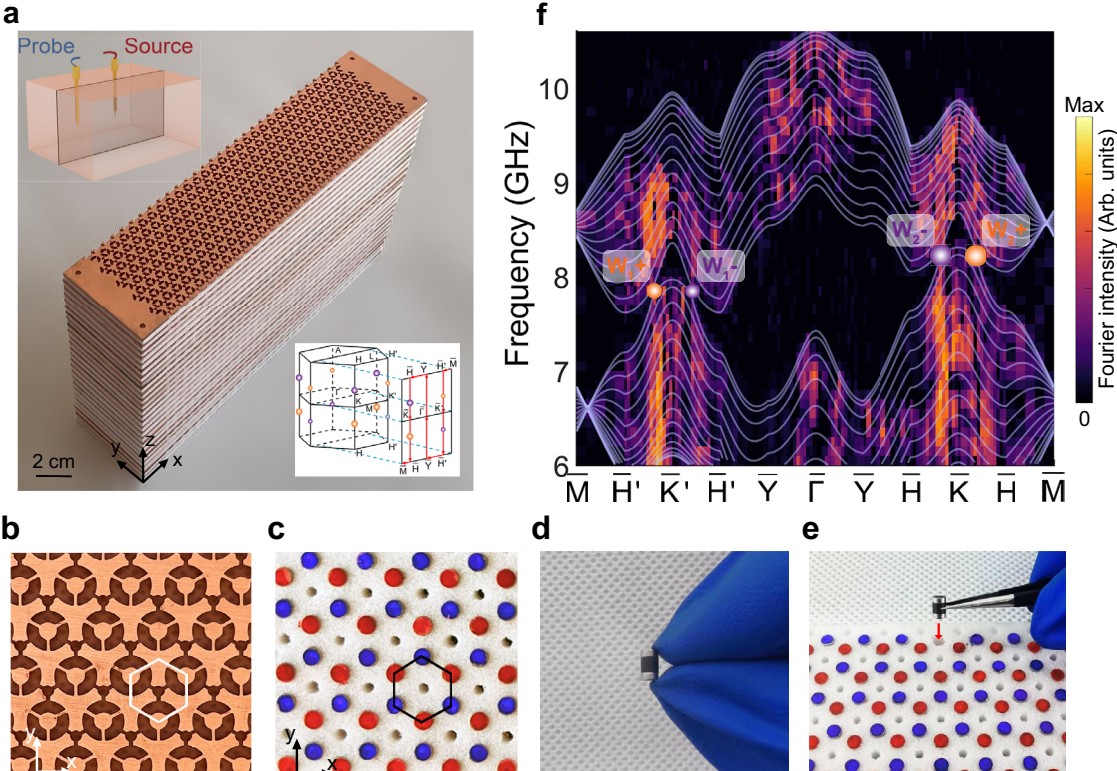

**Fig. 2 | Experimental demonstration of a 3D magnetic Weyl photonic crystal.**
**a** Photograph of the fabricated 3D gyromagnetic photonic crystal that consists of 30 layers in the $z$ direction and $30 \times 10$ unit cells in the $x$-$y$ plane. Left upper inset: schematic of the experimental setup for measuring the bulk band structure. Right lower inset: 3D bulk BZ and projected surface BZ with two pairs of WPs.
**b**, **c** Enlarged photographs of the perforated copper plates and dielectric foams, respectively. Red and blue disks denote the magnetized directions of gyromagnetic rods at different sublattice sites. **d** Magnified image of a gyromagnetic rod (black color) sandwiched between two permanent magnets (silver color). **e** Gyromagnetic rods sandwiched between permanent magnets are inserted into the dielectric foam to fix their positions. **f** Measured (background color) and calculated (white solid lines) projected bulk band structures along high-symmetry lines of the projected surface BZ. Orange and purple spheres represent WPs with opposite topological charges of +1 and −1, respectively.

semimetal with frequency-shifted WPs has never been reported previously in any physical system.

## Experimental observation of topological antichiral surface states

Next, we explore the unique topological antichiral surface states supported by this magnetic Weyl photonic crystal. We cover both A-type (010) and B-type ($0\bar{1}0$) surfaces of the experimental sample with copper plates acting as trivial photonic bandgap materials (see the detailed experimental setup in Supplementary Information). We repeat the field mapping measurements with the point source (cyan star) placed at the center of the A-type (Fig. 3a) and B-type (Fig. 3e) surfaces (see Supplementary Information for the measured electric amplitude and phase distributions of the antichiral surface states), respectively, and conduct Fourier transform to the measured surface field distributions. For the A-type and B-type surfaces, we plot the measured iso-frequency contour of the surface states (background color maps) at three different frequencies (two Weyl frequencies at 7.85 GHz and 8.22 GHz and an intermediate frequency at 8.02 GHz) in Fig. 3b–d, f–h, respectively, which exhibit excellent agreement with the calculated iso-frequency contour (white solid lines represent bulk states and green solid lines represent surface states). At the Weyl frequency of 7.85 GHz (8.22 GHz), the measured surface intensity of A-type surface forms a curve known as single open Fermi arc connecting two lower projected WPs $W_{1+}$ and $W_{1-}$ (upper projected WPs $W_{2+}$ and $W_{2-}$), which is the simplest possible Fermi arc configuration, as shown in Fig. 3b (Fig. 3d). While the situation is different for the B-type surface whose measured surface intensity forms two open

Fermi arcs that connect the lower projected WPs $W_{1+}$ and $W_{1-}$ (upper projected WPs $W_{2+}$ and $W_{2-}$) and right bulk states (left bulk states) due to frequency shift of two WPs pairs, as shown in Fig. 3f (Fig. 3h). At the intermediate frequency of 8.02 GHz, for the A-type surface, its measured surface intensity forms a single Fermi loop winding around the surface BZ, as shown in Fig. 3c. While for the B-type surface, its measured surface intensity forms two open Fermi arcs that connect the projections of two bulk states, as shown in Fig. 3g. Hence, we demonstrate that the surface state iso-frequency contours of the A-type surface can evolve from a single open Fermi arc to a single surface Fermi loop and finally to a single open Fermi arc as the frequency changes. While for the B-type surface, its surface state iso-frequency contours keep unchanged with two open Fermi arcs as the frequency changes. The unidirectional propagation and topologically protected robustness of the antichiral surface states have also been numerically studied and the results are shown in Supplementary Information.

## Tilted surface dispersions of topological antichiral surface states

To further demonstrate the antichiral properties of topological surface states in the magnetic Weyl photonic crystal, Fig. 4a–c, d–f present the frequency-dependent surface dispersions within different $k_z$ slices for the A-type and B-type surfaces, respectively, in which the background color maps represent the measured results, the white (green) solid lines represent the simulated bulk (surface) dispersions, and the purple and orange dots represent the projections of WPs with opposite topological charges. We clearly observe the evolution of the Fermi arc

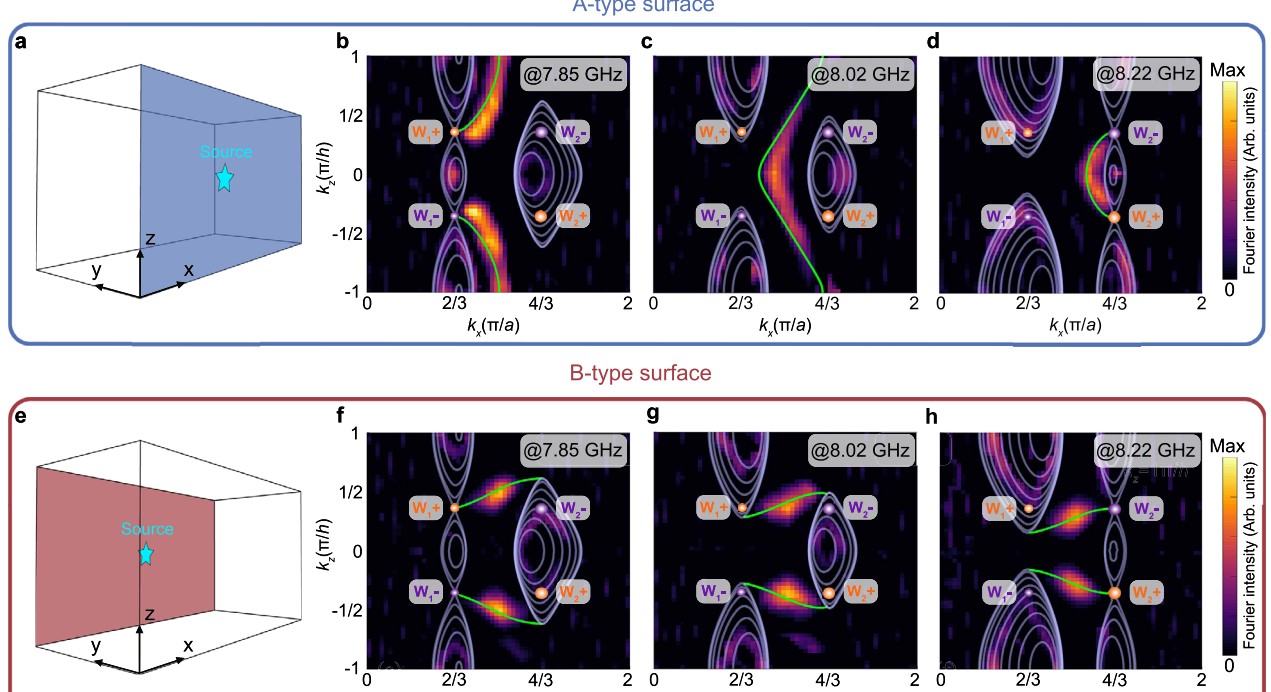

**Fig. 3 | Observation of antichiral surface states in a 3D magnetic Weyl photonic crystal. a, e** Schematics of experimental setup for mapping the electric field distributions of the topological surface state on **a** A-type and **e** B-type surfaces. The point source (cyan star) is placed at the center of the A-type or B-type surface. **b–d** Measured iso-frequency contours of the topological surface states on the A-type surface at **b** 7.85 GHz, **c** 8.02 GHz, and **d** 8.22 GHz. The white (green) lines represent the simulated bulk (surface) dispersions. The orange and purple spheres represent the projections of the frequency-shifted WPs with opposite topological charges. **f–h** Measured iso-frequency contours of the topological surface states on the B-type surface at **f** 7.85 GHz, **g** 8.02 GHz, and **h** 8.22 GHz.

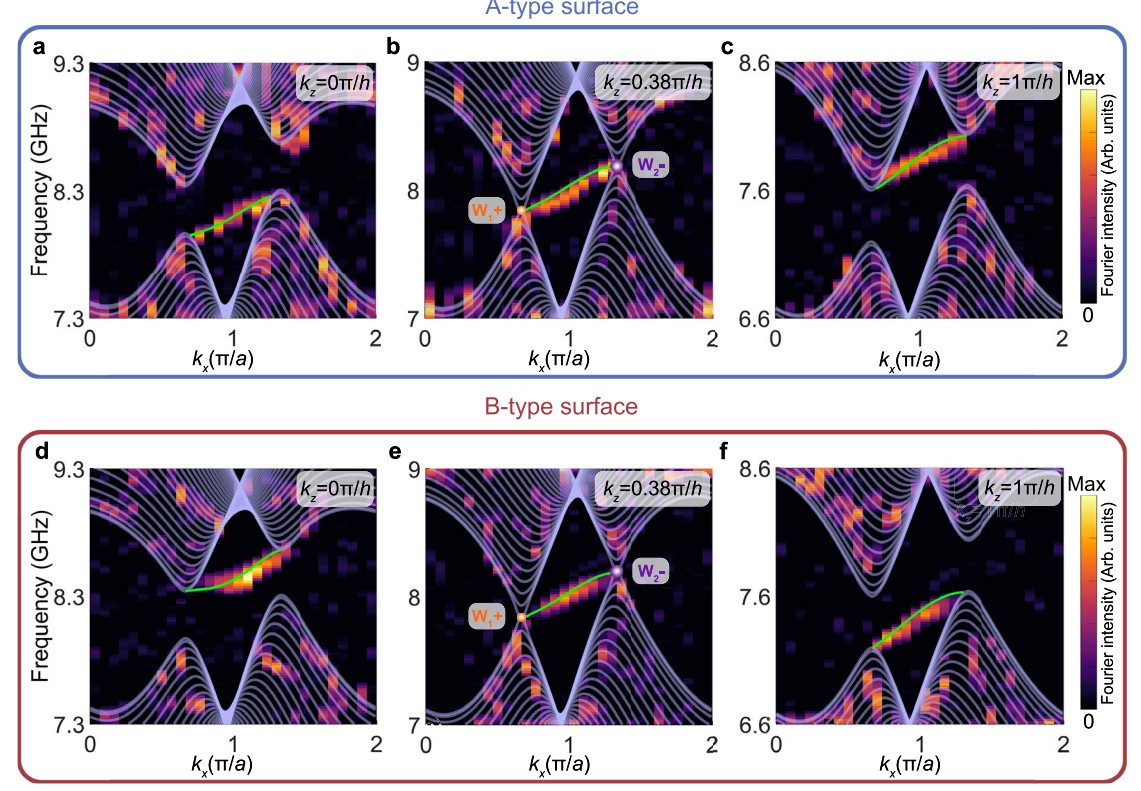

**Fig. 4 | Observation of tilted surface dispersions of antichiral surface states. a–c** Measured surface dispersions of the A-type surface with fixed **a** $k_z = 0\pi/h$, **b** $k_z = 0.38\pi/h$, and **c** $k_z = \pi/h$, respectively. The white (green) lines represent the simulated bulk (surface) dispersions, respectively. The orange and purple spheres represent the projections of the frequency-shifted WPs with opposite topological charges. **d–f** The same as **a–c** but for the B-type surface.

surface state dispersions as a function of $k_z$. At $k_z = 0\pi/h$ and $k_z = 1\pi/h$, for the A-type surface, the projected bulk bandgap opens and the surface states connect the projections of the lower (upper) bulk bands, as shown in Fig. 4a, c, respectively. In contrast, the situations are reversed for the B-type surface, as shown in Fig. 4d, f, respectively. At $k_z = 0.38\pi/h$, for both the A-type (see Fig. 4b) and B-type (see Fig. 4e) surfaces, the bulk bandgap closes and the Fermi arc surface states connect the projections of two frequency-shifted WPs. Remarkably, for both A-type and B-type surfaces with any $k_z$, their topological surface states dispersions always exhibit overall positive slopes along $+k_x$ direction, further confirming the existence of antichiral surface states. The numerically simulated results (white and green solid lines) are consistent with the experimental results (background color maps).

## Discussion

We have theoretically proposed and experimentally observed topological antichiral surface states in a 3D magnetic Weyl photonic crystal which is a photonic analog of a 3D modified Haldane model. By introducing oppositely staggered magnetic fluxes at different sublattice sites of a 3D honeycomb lattice, we experimentally observe two pairs of ideal type-II WPs with a frequency shift, which is a previously unrevealed feature of magnetic Weyl semimetals, besides the robust chiral surface states and the simplest single open Fermi arc. We also provide the experimental demonstration that such frequency-shifted WPs pairs will lead to tilted antichiral surface state dispersions. The counterintuitive antichiral surface states that propagate along the same direction on opposite surfaces of the 3D magnetic Weyl photonic crystals are experimentally verified by measuring their iso-frequency contours and surface state dispersions. As a new type of topologically protected robust one-way transport of electromagnetic waves in 3D space, topological antichiral surface states may lead to potential applications in innovative nonreciprocal photonic devices. Our work expands the landscape of magnetic Weyl semimetals and offers an ideal platform for the investigation of various unconventional physics in magnetic Weyl systems.

## Methods

### Numerical simulation

All numerical results presented in this work are simulated with the RF module of COMSOL Multiphysics. The bulk band structure is simulated using a hexagonal unit cell with periodic boundary conditions in all directions. The A-type and B-type surfaces dispersions are calculated by adopting a $1 \times 20 \times 1$ supercell and applying periodic boundary conditions along $x$ and $z$ directions, and PEC boundary conditions along $y$ direction. Both the copper plates and permanent magnets are modeled as PEC. The fully magnetized gyromagnetic materials have a permeability tensor given by $[\mu_r] = \begin{bmatrix} \mu_m & \pm j\mu_k & 0 \\ \mp j\mu_k & \mu_m & 0 \\ 0 & 0 & 1 \end{bmatrix}$, where $\mu_m = 1 + \omega_m(\omega_0 + i\alpha\omega)/[(\omega_0 + i\alpha\omega)^2 - \omega^2]$, $\mu_k = \omega_m\omega/[(\omega_0 + i\alpha\omega)^2 - \omega^2]$, $\omega_m = \mu_0\gamma M_s$, $\omega_0 = \gamma\mu_0 H_0$, and $\mu_0 H_0$ is the external magnetic field (about 0.115 Tesla) along the $z$ direction, $\gamma = 1.759 \times 10^{11}$ C/kg is the gyromagnetic ratio, $\alpha = 0.0088$ is the damping coefficient, and $\omega$ is the operating frequency[39]. For the 0.115 T external magnetic field, the dispersions of the permeability tensor elements $\mu_m$ and $\mu_k$ are shown in Fig. S1a, b, respectively. It can be seen that the Weyl frequency is far away from the gyromagnetic material's resonant frequency, thus the material dispersion within the operating frequency range is rather weak.

### 3D modified Haldane model

The proposed 3D magnetic Weyl photonic crystal can be qualitatively described by a 3D modified Haldane model. The 3D modified Haldane model is a layer-by-layer stacking of the 2D modified Haldane model[21–25] along the vertical direction with interlayer couplings, as shown in Fig. S2a. Each layer contains two different sublattice sites denoted as $A$ (red spheres) and $B$ (blue spheres) which possess on-site energies of $+M$ and $-M$, respectively. The intralayer couplings contain the nearest-neighbor (NN) coupling $t_1$ (black solid lines) and NNN coupling $t_2 \exp(i\phi)$ (red and blue solid lines), and the NNN coupling $t_2 \exp(i\phi)$ directs inversely in sublattices $A$ and $B$. The interlayer couplings are $t_a$ (purple dashed lines) for the $A$ sublattice site and $t_b$ (orange dashed lines) for the $B$ sublattice site, respectively. The Bloch Hamiltonian of this 3D modified Haldane model is given by $H(\mathbf{k}) = \begin{pmatrix} \Delta^+ & \Phi^- \\ \Phi^+ & \Delta^- \end{pmatrix}$, where $\Delta^\pm = 2t_2 \sum_{n=1,2,3} \cos(\phi - \mathbf{k} \cdot \mathbf{b}_n) \pm M + 2t_{a,b} \cos(k_z h)$, $\Phi^\pm = t_1 \sum_{n=1,2,3} \cos(\mathbf{k} \cdot \mathbf{a}_n) \pm i \sin(\mathbf{k} \cdot \mathbf{a}_n)$, $(\mathbf{a}_1, \mathbf{a}_2, \mathbf{a}_3)$ are the displacements from a $B$ site to its three adjacent $A$ sites, and $\mathbf{b}_1 = \mathbf{a}_2 - \mathbf{a}_3$, $\mathbf{b}_2 = \mathbf{a}_3 - \mathbf{a}_1$, $\mathbf{b}_3 = \mathbf{a}_1 - \mathbf{a}_2$. As shown in Fig. S2c, the calculated upper and lower bulk bands of the 3D modified Haldane model linearly intersect with each other at $k = (4\pi/3, 0, \pm 0.5\pi)$ along the KH line and $k = (-4\pi/3, 0, \pm 0.5\pi)$ along the K′H′ line, forming two pairs of WPs with different energies (marked by purple and orange spheres) in the 3D BZ shown in Fig. S2b. Figure S2d shows the calculated Fermi arc surface states dispersions of A-type (blue color) and B-type (red color) surfaces, respectively. Since two pairs of WPs exhibit a frequency shift, their corresponding Fermi-arc surface states will also be tilted to form antichiral surface states. It can be observed that, for arbitrary $k_z$, the surface state dispersions on A-type (blue curved sheet) and B-type (red curved sheet) surfaces are both tilted and exhibit an overall positive group velocity along $+k_x$ direction, indicating the Weyl surface states on both A-type and B-type surfaces propagate along the same direction.

### Calculation of Weyl point charge in the magnetic Weyl photonic crystal

By tracking the evolution of the Berry phase on a small sphere covering the WPs, we can calculate the topological charge (Chern numbers) of a WP which is a topological invariant of Weyl semimetals. For numerical calculation, the sphere needs to be discretized into a sequence of horizontal loops. We then calculate the Berry phase along the horizontal loops using the Wilson loop: each horizontal loop is divided into $N$ segments, in a small segment from $k_i$ to $k_{i+1}$ ($i$ ranges from 1 to $N$, and $k_{N+1} = k_1$), the Berry phase is given by $W(k_z) = e^{-i\phi(k_z)} = \prod_{i=1}^{N} \langle u_{(n,k_i)}(\mathbf{r}) | u_{(n,k_{i+1})}(\mathbf{r}) \rangle$, where $u_{(n,k_i)}(\mathbf{r})$ is the periodic part of the Bloch eigenstates. The Berry phase along the horizontal loop for a fixed $k_z$ is given by $\phi(k_z) = -\mathrm{Im}[\ln W(k_z)]$. The Chern number $C$ is obtained by counting the winding of $\phi(k_z)$ when $k_z$ goes from the south pole to the north pole of the sphere (i.e., the polar angle $\theta$ in spherical coordinates varies from $-\pi/2$ to $\pi/2$). In Fig. S3, we show the calculated topological charge of the WPs marked by orange and purple spheres along the KH line in Fig. 1e. For the WPs marked by orange sphere, the Berry phase winds one time from $-\pi$ to $+\pi$ with a positive slope, indicating that its topological charge is $+1$, while for the WPs marked by purple sphere, the Berry phase winds one time from $-\pi$ to $+\pi$ with a negative slope, revealing that its topological charge is $-1$. The sign of topological charge depends on the slope of the Berry phase.

### Materials and experimental setups

In the experiment, we adopt commercially available gyromagnetic materials (yttrium iron garnet (YIG) ferrites) and permanent magnets to break the time-reversal symmetry. The radius and height of YIG ferrites are 1.3 mm and 2 mm, respectively. The YIG ferrites have a saturation magnetization $M_s = 1780$ Gauss. Its permittivity and permeability are about $\varepsilon_r = 14.3 + 0.003i$ and

$\mu_r \approx 1$, which are essentially constant at microwave frequencies. The permanent magnets (Sm2Co17) are electroplated by nickel with a thickness of 0.002 mm. One pair of magnets provide an overall uniform external magnetic field of about 0.115 Tesla to magnetize the gyromagnetic rods. The metallic plates are perforated with air holes precisely using a laser cutting technique. To fix the positions of the sandwiched YIG rods and permanent magnets, we adopt the perforated dielectric foam (ROHACELL 31 HF) with relative permittivity 1.04 and loss tangent 0.0025. In the experimental measurements, two microwave dipole antennas function as source and probe are connected to a vector network analyzer (Keysight E5080). By inserting the probe into the air holes one by one and scanning along the z-direction in small steps, we can map the complex electric field distributions on the surfaces or within the bulk of the experimental sample. To obtain the projected bulk and surface band structures, 2D Fourier transformation was performed to the measured complex electric field distributions at each frequency.

## Data availability

Data supporting key conclusions of this work are included within the article and Supplementary information. All data that support the plots within this paper and other findings of this study are available from the corresponding author upon reasonable request.

## Code availability

The code that supports the plots within this paper and other findings of this study is available from the corresponding authors upon reasonable request.

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

## Acknowledgements

Z.G. acknowledges support from the National Natural Science Foundation of China under Grant No. 12104211, Shenzhen Science

and Technology Innovation Commission under Grant No. 20220815111105001, and SUSTech under Grant No. Y01236148 and No. Y01236248. Work at Nanyang Technological University was sponsored by Singapore Ministry of Education Academic Research Fund Tier 3 Grant MOE2016-T3-1-006, and the National Research Foundation Competitive Research Program NRF-CRP23-2019-0007. Work at Zhejiang University was sponsored by the National Natural Science Foundation of China under grant number 62175215 and the Fundamental Research Funds for the Central Universities (2021FZZX001-19).

## Author contributions

Z.G. and G.-G.L. initiated the idea. X.X., L.Y., and G.-G.L. performed the simulations. Z.G., G.-G.L., and B.Z. designed the experiments. Z.G., X.X., B.Y., L.Y., Y.M., Z.-X.Z., J.-M.C., Z.W., and P.Z. fabricated samples. X.X., Z.G., B.Y., L.Y., Y.M., and Z.-X.Z. carried out the measurements. X.X., G.-G.L., Z.G., and B.Z. analyzed data. X.X., Z.G., G.-G.L., and B.Z. wrote the manuscript with input from P.P.S., S.M., Y.Y., and H.C. Z.G., G.-G.L., and B.Z. supervised the project.

## Competing interests

The authors declare no competing interests.
