## [Peer Review File · Nature Communications]

REVIEWER COMMENTS

Reviewer #1 (Remarks to the Author):

The authors theoretically and experimentally demonstrate the antichiral surface states by constructing a three-dimensional modified Haldane model in a magnetic Weyl photonic crystal. The paper is definitely interesting and potentially useful to a wide audience, including researchers working on electronics, photonics and phononics. I thus suggest its acceptance for Nature Communications. But I have some minor questions/comments, which may be clarified prior to its publication.

1. When the antichiral surface states propagate forwards along x direction in x-z plane, will the energy fluxes leak into the air along the +z and -z directions? If so, do you need to cover the upper and lower sides of the sample with metal plates to avoid the leakage? Besides, will the scattering from the metal plates covered in the upper and lower sides of the sample affect the dispersions and transmission behaviors of the antichiral surface states? I suggest that the authors can demonstrate the transmission phenomena of antichiral surface states.
2. The authors mentioned in the ABSTRACT that the antichiral robust propagations was observed, but it seems that there are no related experimental and theoretical results. I suggest that the authors can supplement the measured or simulated results of the robust transmission of antichiral surface states.
3. In the calculation of surface dispersions, are the paired magnets on both sides of the gyromagnetic rod part of the unit cell? In the Supporting Information, some more details about the COMSOL simulations.
4. The authors mentioned "For both the A-type and B-type surfaces, varying k_z causes a topological phase transition (band inversion) of the projected 2D surface bands". I have a minor question, why does the topological phase transition occur with the varying of k_z ?

Reviewer #2 (Remarks to the Author):

The article by Xiang Xi et al. entitled "Topological antichiral surface states in a magnetic Weyl photonic crystal" both theoretically and experimentally reported antichiral surface states in a magnetic Weyl system.

The antichiral edge states usually indicate the edge states that propagate in the same direction at opposite strip edges in 2D. They were proposed by subtly reversing the next-nearest-neighbour (NNN)

couplings in the original 2D Haldane model, where two Dirac points at K and K' locate at different frequencies, and thus the edge states connecting the two frequency-shifted Dirac points are tilted finally. More interestingly, the two edge states are usually degenerate in frequency.

The work for the first time reports the antichiral surface states in a cleverly designed 3D magnetic Weyl semimetal. The theoretical proposal is interesting and the experimental measurements are very clear. All results are convincing. Therefore, the work can be recommended for publication on Nature Communications after the following comments and suggestions are well addressed.

1. Why do the authors call the Weyl points “ideal”? i.e. “3D bulk Brillouin zones with ideal type II WPs” in the caption of Fig. 2. The authors may need to explain this.

2. Why are the two edge/surface states degenerate at the k_z cut (see Fig. 1h)? Is there some underlying symmetry protection? In other words, at the fixed k_z (i.e. $k_z=0.38\pi/h$), the two surface states exactly cross (see Fig. S1b). Why?

The sentence “And the Fermi arc surface state dispersions of A-type (blue dashed line) and B-type (red dashed line) surfaces degenerate...” seems to miss “are”.

3. The schematic view in Fig. 1a seems like two guiding modes rather than the two opposite surface modes.

4. In Fig.2, if a skyline (i.e. the boundary of the bulk states in Fig. 2g) is plotted in Fig. 2f, it would be better for the comparison.

5. Fig. S1b is helpful to understand both the Fermi arcs and the anti-chiral surface states. As anti-chiral surface states are defined in the k_x - f view, while Fermi arcs are given in the k_x - k_z view, Fig. S1b just provides a global view. It would be better if there is a global plot relating to Fig. 3 and Fig. 4.

Reviewer #3 (Remarks to the Author):

The Authors present the theoretical study, design and experimental demonstration of a 3D magnetic 3D Weyl photonic crystal consists of gyromagnetic cylinders with opposite magnetization in different triangular sublattices of a 3D honeycomb lattice that sustains topological antichiral surface states. For the theoretical analysis they employ a properly 3d modified Haldane model and they calculate the topological charge by counting the winding of the Berry phase. For the experimental demonstration they use a system of probe/source antennas with a network analyzer and they measure the field distribution

and subsequently calculate the dispersion diagrams. The work is interesting and timely and the manuscript reads well. Simulations and experiments stand in good agreement. Before suggesting for publication certain issues need to be addressed. My comments are:

1. Since the prominent topological feature is the unidirectional propagation, did the authors considered performing the characterization of the unidirectional propagation of the states in the system? How would the excitation and unidirectionality would be proven in an full wave simulations? Would an experiment demonstration be feasible? If yes how?
2. The Authors investigate the dispersion characteristics in the xz plane. Could the Authors comment on the selection of the y plane (is it in the middle of the unit cell?) and k_y in their investigation?
3. Could the Authors provide a reference regarding the details of the YIG material selected? Also there seems to be an inconsistency in the formulation and the materials set up parameters selected regarding the YIG material. Which is the YIG material resonance for the parameters selected?
4. Is the dispersion of the YIG material taken into account in the numerical simulation of the band structure? If yes how is this incorporated in the Cosmol routine?
5. Could the Auhtors provide the raw data of the E-field distribution before the FFTs?

Response Letter to Referees

We are grateful for the constructive comments on the manuscript (NCOMMS-22-39104-T) from three referees, which have guided us in significantly improving the paper.

In the text below, referee comments are quoted in *italics* and followed by our detailed response. We have also revised the Manuscript and Supplementary Materials based on the referee's comments, and these updates are highlighted in blue and by a vertical red line in the left margin in those files. In the text below, the references to these updates are highlighted in a similar way.

GENERAL COMMENTS FROM REFEREE #1:

The authors theoretically and experimentally demonstrate the antichiral surface states by constructing a three-dimensional modified Haldane model in a magnetic Weyl photonic crystal. The paper is definitely interesting and potentially useful to a wide audience, including researchers working on electronics, photonics and phononics. I thus suggest its acceptance for Nature Communications. But I have some minor questions/comments, which may be clarified prior to its publication.

Response from Authors:

We thank Referee #1 for considering our work “*definitely interesting and potentially useful to a wide audience*” and suggesting “*acceptance for Nature Communications*”. In the following, we will address the referee's specific comments point-by-point.

SPECIFIC COMMENTS FROM REFEREE #1:

Referee #1 – Comment 1:

When the antichiral surface states propagate forwards along x direction in x - z plane, will the energy fluxes leak into the air along the $+z$ and $-z$ directions? If so, do you need to cover the upper and lower sides of the sample with metal plates to avoid the leakage? Besides, will the scattering from the metal plates covered in the upper and lower sides of the sample affect the dispersions and transmission behaviors of the antichiral surface states? I suggest that the authors can demonstrate the transmission phenomena of antichiral surface states.

Response from Authors:

We thank Referee #1 for his/her careful reading and useful suggestions.

The energy fluxes of the antichiral surface states will *not* leak into the air along the $+z$ and $-z$ directions. To verify this argument, we have demonstrated the transmission phenomena of antichiral surface states from two aspects: full-wave simulation in real space and phase-matching analysis in reciprocal space.

Full-wave simulation in real space:

As plotted in Fig. R1a, we construct a finite structure with the A-type surface covered with copper claddings and the upper and lower sides surrounded by air. We put an electric dipole source near the A-type surface and perform the full-wave simulation using the finite-element-method solver (COMSOL Multiphysics RF module). When the source oscillates at 8.02 GHz (between the frequencies of two pairs of Weyl points), the surface states will be excited and propagate rightwards, as plotted in Fig. R1a. Clearly, when reaching the upper or lower sides, the surface states will be totally reflected back to the A-type surface, and cannot leak into the surrounding air along the $+z$ and $-z$ directions. Besides, on the upper and lower surfaces, the

electric fields decay exponentially along the y direction, manifesting that the surface state cannot couple into the top and bottom surfaces. Similar results have also been found for the simulated surface states on the B-type surface, as shown in Fig. R1d.

Phase-matching analysis in reciprocal space

Here, we have analyzed the phase-matching condition at the interface between the A-type surface and the surrounding air. As shown in the left panel of Fig. R1b, we have plotted the isofrequency contour of A-type surface states at 8.02 GHz as red lines, and its dispersion in the air as a blue circle. Clearly, as required by the phase-matching condition, there is no mode that can couple from the A-type surface states into the air cladding. Therefore, the surface states on the A-type surface will be totally reflected, as indicated in the right panel of Fig. R1b. The situation is similar for the excited surface states on the B-type surface, as shown in Fig. R1e.

In summary, the surface states will *not* leak into the air along the $+z$ and $-z$ directions, as manifested by full-wave simulation in real space and phase-matching analysis in reciprocal space. Besides, we apply spatial Fourier transformation to the simulated electric field distributions of the A-type (Fig. R1a) and B-type (Fig. R1d) surface states, and obtain the results in Fig. R1c and Fig. R1f, respectively. We notice that the transformed Fourier intensities (background colors) are consistent with the calculated iso-frequency contours (green lines), manifesting that the total reflection at the terminal interfaces between the sample and the air claddings does not affect the dispersions of the antichiral surface states. Note that although our simulation and analysis are based on one particular surface state frequency (8.02 GHz), the analysis method and conclusion are applicable to all surface state frequencies.

Fig. R1 | Transmission phenomena of antichiral surface states. **a, d** Simulated electric field distributions of the topological surface states on **a** A-type and **d** B-type surfaces. The exponential decay of electric fields on the top and bottom surfaces of the magnetic Weyl photonic crystal indicates that the top and bottom surfaces do not support any surface states. **b, e** The phase-matching condition in the reciprocal space and the total reflection in the real space for **b** A-type and **e** B-type surfaces. **c, f** Transformed Fourier intensities of simulated fields in **a** and **d** (background colors) and iso-frequency contours for **c** A-type and **f** B-type surface states (green lines), respectively.

Referee #1 -- Comment 2:

The authors mentioned in the ABSTRACT that the antichiral robust propagations was observed, but it seems that there are no related experimental and theoretical results. I suggest that the authors can supplement the measured or simulated results of the robust transmission of antichiral surface states.

Response from Authors:

We thank Reviewer 1# for this constructive suggestion.

During the revision process, we numerically study the robustness of antichiral surface states. As shown in Fig. R2, we insert a metallic obstacle (yellow rod) in the path of the surface states on both A-type and B-type surfaces and then perform the full-wave simulation. As can be seen, the surface states on both A-type and B-type surfaces can bypass the obstacle and continue to propagate with negligible reflections, verifying the robustness of the antichiral surface states.

Fig. R2 | Robustness of antichiral surface states. **a, b** Simulated electric field distributions of the topological surface states at 8.02 GHz on **a** A-type and **b** B-type surfaces with a copper pillar (yellow rod) inserted into the photonic crystal as a metallic obstacle.

The detailed analysis has been discussed in the supplementary materials on Page 15. Fig. R2 has been adopted as Fig. S10 in the revised supplementary materials.

Referee #1 -- Comment 3:

In the calculation of surface dispersions, are the paired magnets on both sides of the gyromagnetic rod part of the unit cell? In the Supporting Information, some more details about the COMSOL simulations.

Response from Authors:

Yes, in all numerical simulations, including the band structures and field distributions, the two pairs of magnets on both sides of the gyromagnetic rod have always been part of the unit cell. The numerical simulation models are constructed using only gyromagnetic rods and surrounding air regions as materials, while both the perforated copper plates and magnets are modeled as perfect electric conductor boundaries due to their high conductivity. As shown in Fig. R3, we first choose a hexagonal unit cell (Fig. R3a) which consists of gyromagnetic rods (red and blue colors) and permanent magnets (gray color) stacked on both sides of the perforated copper plates (yellow color). For simplicity, we replace the complementary region of copper plates and permanent magnets with air (gray color in Fig. R3b) and treat copper plates and permanent magnets as perfect electric conductor (PEC) boundary conditions (see Fig. R3c). Finally, we apply periodic boundary conditions (Fig. R3d) to the outmost boundaries to calculate the bulk band structure of the 3D magnetic Weyl photonic crystals. In the

calculation of surface state dispersions, we construct a 1×20 supercell and apply periodic boundary conditions along the x and z directions, and perfect electric conductor boundaries in the y direction.

Fig. R3 | Details of modeling the 3D magnetic Weyl photonic crystal in COMSOL simulations. a Original unit cell with the real structure. **b** The simplified unit cell for calculation in COMSOL simulations. **c** Perfect electric conductor boundary conditions and **d** periodic boundary conditions for the calculated unit cell.

Besides, the detailed analysis has been discussed in the supplementary materials on Page 7. Fig. R3 has been adopted as Fig. S5 in the revised supplementary materials.

Referee #1 -- Comment 4:

The authors mentioned “For both the A-type and B-type surfaces, varying k_z causes a topological phase transition (band inversion) of the projected 2D surface bands”. I have a minor question, why does the topological phase transition occur with the varying of k_z ?

Response from Authors:

We thank Referee #1 for this insightful question. The varying of k_z will cause the inversion of Berry curvature and valley Chern number of the 3D modified Haldane model, thus leading to the topological phase transition. To make this point clear, we have added the following discussions on Pages 8 and 9 in the revised supplementary materials:

“Topological phase transition and Berry curvature of three-dimensional (3D) modified Haldane model. For each fixed k_z , the Berry curvature can be defined in the first two-dimensional (2D) Brillouin zone (BZ) at the $k_x - k_y$ plane. In the case of a 2D honeycomb lattice, we can choose the rhombus as the first BZ, as shown in Fig. S6a, which is defined by the reciprocal lattice vectors and discretizes the unit cell in k_i and k_j directions [Adv. Quantum Technol. 3, 1900117 (2020)]. The Berry curvature can be calculated for each plaquette of the discretization using the four-point formula: for isolated bands, the integration of Berry curvature around a plaquette is given by,

$$\phi(k) = \iint \Omega(k) ds = -\text{Im} \log[\langle \mu_{k_1}(\mathbf{r}) | \mu_{k_2}(\mathbf{r}) \rangle \langle \mu_{k_2}(\mathbf{r}) | \mu_{k_3}(\mathbf{r}) \rangle \langle \mu_{k_3}(\mathbf{r}) | \mu_{k_4}(\mathbf{r}) \rangle \langle \mu_{k_4}(\mathbf{r}) | \mu_{k_1}(\mathbf{r}) \rangle],$$

where $\mu_{k_i}(\mathbf{r})$ are the periodic functions at each corner of each plaquette. To demonstrate the topological phase transition, we first compute the Berry curvature on both sides of the WPs ($k_z = \pm 0.5\pi$), i.e., at $k_z = \pm 0.45\pi$ [see Fig. S6b] and $k_z = \pm 0.55\pi$ [see Fig. S6c], respectively. As shown in Fig. S6b, the integration of Berry curvature in the first BZ is zero, but the integration of Berry curvature in the half BZ, the valley-dependent topological index, is non-zero, indicating that the band is topologically non-trivial. The valley Chern number for $k_z = \pm 0.45\pi$ is $C_v = C_K - C_{K'} = (0.47) - (-0.47) = +0.94$. By contrast, the situation reverses for $k_z = \pm 0.55\pi$. As shown in Fig. S6c, the valley Chern number is $C_v = C_K - C_{K'} = (-0.47) - (0.47) = -0.94$. Although the valley Chern number is not a well-defined integer, i.e., $|C_v| < 1$, the difference in the sign of valley-dependent topological index still ensures that they are topological valley phases [Nat. Commun. 10, 872 (2019)]. Fig. S6d plots the valley Chern number with the varying of k_z . It can be seen that when k_z crosses the projections of WPs ($k_z = \pm 0.5\pi$), the valley Chern number will change from +1 (-1) to -1 (+1). In general, the band structure with positive valley Chern number exhibits different topological properties with that with negative valley Chern number, indicating that topological phase transition occurs with the varying of k_z .”

Fig. S6 | Topological phase transition and Berry curvature of 3D modified Haldane model. a Discretization of the first BZ of honeycomb lattice. **b, c** Berry curvature distributions of 3D modified Haldane model in $k_x - k_y$ plane with **b** $k_z = \pm 0.45\pi$ and **c** $k_z = \pm 0.55\pi$. **d** The valley Chern number with the varying of k_z .

GENERAL COMMENTS FROM REFEREE #2:

The article by Xiang Xi et al. entitled "Topological antichiral surface states in a magnetic Weyl photonic crystal" both theoretically and experimentally reported antichiral surface states in a magnetic Weyl system.

The antichiral edge states usually indicate the edge states that propagate in the same direction at opposite strip edges in 2D. They were proposed by subtly reversing the next-nearest-neighbour (NNN) couplings in the original 2D Haldane model, where two Dirac points at K and K' locate at different frequencies, and thus the edge states connecting the two frequency-

shifted Dirac points are tilted finally. More interestingly, the two edge states are usually degenerate in frequency.

The work for the first time reports the antichiral surface states in a cleverly designed 3D magnetic Weyl semimetal. The theoretical proposal is interesting and the experimental measurements are very clear. All results are convincing. Therefore, the work can be recommended for publication on Nature Communications after the following comments and suggestions are well addressed.

Response from Authors:

We thank Referee #2's appreciation that our work "for the first time reports the antichiral surface states in a cleverly designed 3D magnetic Weyl semimetal" and "The theoretical proposal is interesting and the experimental measurements are very clear. All results are convincing". We also thank his/her recommendation for publication on Nature Communications. In the following, we will address his/her comments and suggestions with point-by-point responses.

SPECIFIC COMMENTS FROM REFEREE #2:

Referee #2 -- Comment 1:

Why do the authors call the Weyl points "ideal"? i.e. "3D bulk Brillouin zones with ideal type II WPs" in the caption of Fig. 2. The authors may need to explain this.

Response from Authors:

For the two pairs of type-II WPs along the HK (H'K') high-symmetry lines, there exist no additional trivial bands at the same frequency, hence we dub them as ideal type-II WPs. This definition of ideal type-I and type-II WPs has also been widely adopted by many previous works such as [*Science* 359, 1013 (2018); *Phys. Rev. Lett.* 125, 143001 (2020); *Natl. Sci. Rev.* 8, nwaal92 (2020)].

In the revised manuscript, we have clarified this point on Page 4, starting from line 4. It read as:

"since there exist no additional trivial bands at the same Weyl frequency"

Besides, three new references about the naming of ideal type-I and type-II WPs are cited on Page 13 in the revised manuscript:

36. B. Yang, Q. Guo, B. Tremain, R. Liu, L. E. Barr, Q. Yan, W. Gao, H. Liu, Y. Xiang, J. Chen, C. Fang, A. Hibbins, L. Lu, and S. Zhang, Ideal Weyl points and helicoid surface states in artificial photonic crystal structures, *Science* **359**, 1013 (2018).
37. Y. Yang, Z. Gao, X. Feng, Y. X. Huang, P. Zhou, S. A. Yang, Y. Chong, and B. Zhang, Ideal unconventional Weyl point in a chiral photonic metamaterial, *Phys. Rev. Lett.* **125**, 143001 (2020).
38. R. Li, B. Lv, H. Tao, J. Shi, Y. Chong, B. Zhang, and H. Chen, Ideal type-II Weyl points in topological circuits, *Natl. Sci. Rev.* **8**, nwaal92 (2020)."

Referee #2 -- Comment 2:

Why are the two edge/surface states degenerate at the k_z cut (see Fig. 1h)? Is there some underlying symmetry protection? In other words, at the fixed k_z (i.e. $k_z=0.38\pi/h$), the two surface states exactly cross (see Fig. S1b). Why? The sentence "And the Fermi arc surface

state dispersions of A-type (blue dashed line) and B-type (red dashed line) surfaces degenerate...” seems to miss “are”.

Response from Authors:

We thank Referee #2 for this constructive and insightful question. The reason for the degeneracy of two antichiral surface states at fixed $k_z = 0.38\pi/h$ is because of the mirror symmetry in the 2D modified Haldane model. In both numerical simulations and tight-binding models we can observe the degeneracy. Here we adopt tight-binding models to explain the reasons. For convenience, we rewrite the 3D modified Haldane model as

$$H_{3D} = d_x \sigma_x + d_y \sigma_y + d_z \sigma_z + d_0 I, \quad (1)$$

where

$$d_x = t_1 \sum_{n=1,2,3} \cos(\mathbf{k} \cdot \mathbf{a}_n), \quad (2)$$

$$d_y = t_1 \sum_{n=1,2,3} \sin(\mathbf{k} \cdot \mathbf{a}_n), \quad (3)$$

$$d_z = M + (t_a - t_b) \cos(k_z h), \quad (4)$$

$$d_0 = 2t_2 \sum_{i=1,2,3} \cos(\phi - \mathbf{k} \cdot \mathbf{b}_i) + (t_a + t_b) \cos(k_z h). \quad (5)$$

σ are the Pauli matrices and I is the identity matrix. The eigenvalues of the Hamiltonian (1) are $E_{\pm} = d_0 \pm \sqrt{d_x^2 + d_y^2 + d_z^2}$. When we fix k_z at the projections of WPs with $d_z = 0$, this model is simplified to a 2D modified Haldane model

$$H_{2D} = d_x \sigma_x + d_y \sigma_y + d_0 I. \quad (6)$$

Note that $(t_a + t_b) \cos(k_z h)$ term in (5) only shifts the energy of the band structure. In the 2D modified Haldane model, two antichiral edge states on opposite stripe edges are always degenerate because the d_0 term is proportional to the identity matrix. Compared to the pristine graphene model which has two degenerate zero edge modes on opposite stripe edges, the d_0 term does not change the degeneration of the wave functions but changes their energies which makes the edge dispersions tilt [*Phys. Rev. Lett.* 120, 086603 (2018)]. The degeneration can also be further verified by the mirror symmetry of 2D modified Haldane model in a finite lattice ($M_x H M_x^\dagger = H$). Suppose that $|\psi_1\rangle = \psi_0 e^{-\xi|y+y_0|}$ is an eigenvector of the edge state with eigenvalue E ; it also exists the other eigenvector $|\psi_2\rangle = M_x |\psi_1\rangle = \psi_0 e^{-\xi|y-y_0|}$ with the same eigenvalue E , since $H(M_x |\psi\rangle) = M_x H |\psi\rangle = M_x (E |\psi\rangle) = E (M_x |\psi\rangle)$, as shown in Fig. R4. Thus $|\psi_1\rangle$ and $|\psi_2\rangle$ must be two different but degenerate antichiral edge states.

Fig. R4 | Schematic view of the 2D modified Haldane model, where $|\psi_1\rangle$ and $|\psi_2\rangle$ are two different but degenerate antichiral edge states due to the mirror symmetry.

In the revised manuscript, we have clarified this point on Page 5, starting from line 7, in the revised manuscript with following sentence:

“The reason for the degeneracy of two antichiral surface states at fixed $k_z = 0.38\pi/h$ is because of the mirror symmetry in the 2D modified Haldane model (see detailed analysis in Supplementary Materials).”

and Fig. R4 has been adopted as Fig. S12 with related discussions on Pages 17-18 in the revised supplementary materials.

Moreover, we have added “are” to the sentence “And the Fermi arc surface state dispersions of A-type (blue dashed line) and B-type (red dashed line) surfaces degenerate...” in the revised manuscript.

Referee #2 -- Comment 3:

The schematic view in Fig. 1a seems like two guiding modes rather than the two opposite surface modes.

Response from Authors:

We thank Referee #2 for his/her careful reading and suggestion. To avoid confusion, we have redrawn the schematic view, as shown in Fig. R5:

Fig. R5 | Conceptual illustration of antichiral surface states (red and blue arrows) that propagate in the same direction on opposite surfaces of a 3D gyromagnetic Weyl photonic crystal.

Besides, Fig. R5 has been adopted as Fig. 1a on Page 14 in the revised manuscript.

Referee #2 -- Comment 4:

In Fig.2, if a skyline (i.e. the boundary of the bulk states in Fig. 2g) is plotted in Fig. 2f, it would be better for the comparison.

Response from Authors:

We thank Referee #2 for this good suggestion. As shown in Fig. R6, we have replotted a skyline of numerical bulk states (white solid lines) overlapped on the color map of experimental results for better comparison.

Fig. R6 | Measured (background color map) and numerically (white solid lines) calculated projected bulk band structures along high-symmetry lines of the projected surface BZ. Orange and purple spheres represent WPs with opposite topological charges of +1 and -1, respectively.

Besides, we have replaced Fig. 2f with Fig. R6 on Page 15 in the revised manuscript.

Referee #2 -- Comment 5:

Fig. S1b is helpful to understand both the Fermi arcs and the anti-chiral surface states. As anti-chiral surface states are defined in the kx - f view, while Fermi arcs are given in the kx - kz view,

Fig. S1b just provides a global view. It would be better if there is a global plot relating to Fig. 3 and Fig. 4.

Response from Authors:

We thank Reviewer #2 for his/her constructive suggestion. To provide a global plot relating to Fig. 3 and Fig. 4, we have numerically studied antichiral surface states and Fermi arcs based on the 3D modified Haldane model and added the following discussions on Pages 5 and 6 in the revised supplementary materials:

“Iso-energy contours and tilted surface dispersions of antichiral surface states in 3D modified Haldane model. Fig. S4a-c and Fig. S4d-f show the iso-energy contours of the antichiral surface states calculated by the 3D modified Haldane model on the A-type and B-type surfaces, respectively. For the A-type surface, as shown in Fig. S4a-c, its surface state iso-energy contours evolve from a single open Fermi arc connecting two projected WPs (Fig. S4a) to a single surface Fermi loop winding around the surface Brillouin zone (Fig. S4b), and finally return to a single open Fermi arc connecting the other two projected WPs (Fig. S4c). For the B-type surface, as shown in Fig. S4d-f, its iso-energy contours consist of two open Fermi arcs and keep unchanged with the variation of energies. Fig. S4g-i show the antichiral surface state dispersions at three different k_z slices for A-type (blue solid line) and B-type (red solid line) surfaces. For the A-type surface, as shown in Fig. S4g (Fig. S4i), the bulk bandgap opens and the surface states connect the projections of the lower (upper) bulk bands at $k_z = 0\pi$ ($k_z = \pi$), respectively. By contrast, the situation reverses for the B-type surface. At $k_z = 0.5\pi$, as shown in Fig. S4h, the bulk bandgap closes and the Weyl surface states connect the projections of two energy-shifted WPs for both the A-type and B-type surfaces. The numerical results calculated by the 3D modified Haldane model match well with our simulation and experimental results shown in Fig. 3 and Fig. 4 in the main text.”

Besides, to make the results calculated by the 3D modified Haldane model match well with our simulation results calculated by COMSOL Multiphysics, we refine the couplings in 3D modified Haldane model on Page 3 in the revised supplementary materials:

“The 3D modified Haldane model with nearest-neighbor (NN) coupling ($t_1 = 18$), next-nearest-neighbor (NNN) coupling ($t_2 = 1$ and $\phi = -\pi/3$), and interlayer couplings ($t_a = 1$ and $t_b = 10$), respectively.”

Fig. S4 | Iso-energy contours and tilted surface dispersions of antichiral surface states in 3D modified Haldane model. Calculated iso-energy contours of the topological surface states on the **a-c** A-type surface and **d-e** B-type surface at **a, d** $E = -6.23$, **b, e** $E = 0$ and **c, f** $E = 2.73$, respectively. The cyan (green) lines represent the calculated bulk (surface) dispersions, respectively. The orange and purple spheres represent the projections of the energy-shifted WPs with opposite topological charges. **g-i** Calculated surface dispersions for fixed values of **g** $k_z = 0\pi$, **h** $k_z = 0.5\pi$ and **i** $k_z = 1\pi$, respectively. The blue (red) lines indicate the A-type (B-type) surface state dispersions, respectively, and cyan regions represent the projected bulk states.

GENERAL COMMENTS FROM REFEREE #3:

The Authors present the theoretical study, design and experimental demonstration of a 3D magnetic 3D Weyl photonic crystal consists of gyromagnetic cylinders with opposite magnetization in different triangular sublattices of a 3D honeycomb lattice that sustains topological antichiral surface states. For the theoretical analysis they employ a properly 3d modified Haldane model and they calculate the topological charge by counting the winding of the Berry phase. For the experimental demonstration they use a system of probe/source

antennas with a network analyzer and they measure the field distribution and subsequently calculate the dispersion diagrams. The work is interesting and timely and the manuscript reads well. Simulations and experiments stand in good agreement. Before suggesting for publication certain issues need to be addressed.

Response from Authors:

We thank Referee #3 for considering our work “*interesting*,” “*timely*” and “*reads well*”. In the following, we will address his/her specific comments point by point.

SPECIFIC COMMENTS FROM REFEREE #3:

Referee #3 – Comment 1:

Since the prominent topological feature is the unidirectional propagation, did the authors considered performing the characterization of the unidirectional propagation of the states in the system? How would the excitation and unidirectionality would be proven in an full wave simulations? Would an experiment demonstration be feasible? If yes how?

Response from Authors:

We thank Referee #3 for his/her constructive questions on the unidirectionality of antichiral surface states, we have considered the characterization of the unidirectional propagation of the antichiral surface states. In the numerical simulations, we put a point source at the center of A-type and B-type surfaces to excite the antichiral surface states. From the simulated electric field distributions of the antichiral surface states, we find both of them propagate rightward, thus proving the unidirectionality of the antichiral surface states. In the experiments, we have conducted near-field imaging measurements to demonstrate the unidirectional propagation of the antichiral surface states. To make this point clear, we have added the following discussions on Page 15 of the revised supplementary materials:

“**Unidirectional propagation of antichiral surface states.** One of the most interesting properties of magnetic Weyl photonic crystals is the unidirectional propagation of topological surface states due to the time-reversal symmetry breaking. To characterize this unique property, we first perform full-wave simulations on a finite 3D magnetic Weyl photonic crystal. An electric dipole source (cyan star) is placed at the center of A-type and B-type surfaces to excite the surface states, as shown in Fig. S9a-b. For both A-type and B-type surfaces, we observe that the excited antichiral surface states always propagate rightward along the $+x$ direction, unambiguously verifying the unidirectional propagation of antichiral surface states. We then perform electromagnetic near-field imaging measurements to probe the unidirectional propagation characteristic of the antichiral surface states. We cover the frontal (A-type) and back (B-type) surfaces of the experimental sample with copper claddings, and all other surfaces with microwave absorbers. A microwave dipole antenna source (cyan star) is placed at the center of the front and back surfaces to excite the surface states. Fig. S9c-d show the measured electric field distributions of the antichiral surface states on A-type and B-type surfaces, revealing that topological antichiral surface states propagate unidirectionally along $+x$ direction for both surfaces.”

Fig. S9 | Unidirectional propagation of antichiral surface states. **a, b** Simulated and **c, d** measured electric field distributions of antichiral surface states at 8.02 GHz on the **a, c** A-type and **b, d** B-type surfaces, respectively, revealing the unique antichiral and unidirectional propagation properties of the antichiral surface states on opposite surfaces of the magnetic Weyl photonic crystal.

We also highlight this point on Page 6, starting from line 25, in the revised manuscript with following sentences:

“The unidirectional propagation and topologically protected robustness of the antichiral surface states have also been numerically studied and the results are shown in Supplementary Materials.”

Referee #3 -- Comment 2:

The Authors investigate the dispersion characteristics in the xz plane. Could the Authors comment on the selection of the y plane (is it in the middle of the unit cell?) and k_y in their investigation?

Response from Authors:

We thank Referee #3 for his/her comment on the selection of the y plane and k_y . Indeed, the selection of different y planes will significantly modify the surface state dispersions in the xz plane, while k_y has no influence on the surface state dispersions in the xz plane. To make this point clear, we have added the following discussions on Pages 12 and 13 in the revised supplementary materials:

“**Surface state dispersions with two different basic boundary selections.** Different from conventional topological surface states protected by Chern number, the antichiral surface states protected by the valley Chern number depend significantly on the selection of boundary since the nonzero valley Chern number only ensures the existence of surface states but does not guarantee their shapes and slopes. Consequently, the surface state dispersions will be modified by cutting the boundaries at different y planes. As shown in Fig. S8a-b, there exist two basic boundary selections, i.e., one is at the lattice point of the unit cell (Fig. S8a) and the other is at

the middle of the unit cell (Fig. S8b). We have studied the former case in detail in the main text. Here, we focus on the latter case with the boundary located at the middle of the unit cell (Fig. S8b). The calculated surface state dispersions are shown in Fig. S8c-e. For the A-type surface (blue solid line), as shown in Fig. S8c and Fig. S8e, the bulk bandgap opens and the surface state connects the projections of the lower bulk bands at $k_z = 0\pi/h$, while the surface state disappears at $k_z = 1\pi/h$. The situation reverses for the B-type surface (red solid line), which disappears at $k_z = 0\pi/h$ but connects the projections of the upper bulk bands at $k_z = 1\pi/h$. At $k_z = 0.38\pi/h$, as shown in Fig. S8d, the bulk bandgap closes and the Weyl surface states (red and blue solid lines) connect the projections of two frequency-shifted WPs. For the A-type surface, the Weyl surface state (blue solid line) connects the projections of two frequency-shifted WPs through the BZ boundary ($k_x = 1\pi/a$), while for the B-type surface, the Weyl surface state (red solid line) connects the projections of two frequency-shifted WPs through the BZ center ($k_x = 0\pi/a$). Noted that all these surface states own both positive and negative group velocities at the same frequency along k_x direction, indicating that the unidirectional propagation characteristic of the surface states disappears when we cut the boundary at the middle of the unit cell.”

Fig. S8 | Surface state dispersions for two different basic boundary selections. **a, b** Schematics of the boundary for the ribbon cutting at **a** the lattice point of the unit cell and **b** the middle of the unit cell. Red dashed rectangles represent the top view of the supercells with different boundary selections used to calculate the surface state dispersions. **c-e** Simulated surface states dispersions for fixed values of **c** $k_z = 0\pi/h$, **d** $k_z = 0.38\pi/h$ and **e** $k_z = 1\pi/h$ with the boundary cut at the middle of the unit cell shown in **b**. Blue (red) solid lines indicate the A-type (B-type) surface state dispersion, respectively, and cyan regions represent the projected bulk states.

And we also highlight the point on Page 4, starting from line 18, in the revised manuscript:

“Note that the surface states dispersions of A-type and B-type surfaces in the xz plane depend significantly on the cutting position along y direction (Two different boundary selections are analyzed in detail in Supplementary Materials).”

Note that since the surface states dispersions are calculated in the xz plane, therefore, k_y is always fixed as zero.

Referee #3 -- Comment 3:

Could the Authors provide a reference regarding the details of the YIG material selected? Also there seems to be an inconsistency in the formulation and the materials set up parameters selected regarding the YIG material. Which is the YIG material resonance for the parameters selected?

Response from Authors:

We thank Referee #3 for his/her comment on the details of the yttrium iron garnet (YIG) material selected. The formulations of the YIG material used in this paper are given by the textbook [D. M. Pozar, *Microwave Engineering* (Wiley, 2012)]. The corresponding description can also be found on Page 8, starting from line 4, in the revised manuscript:

“The relative permeability tensor of the gyromagnetic materials has the form

$$[\mu_r] = \begin{bmatrix} \mu_m & \pm j\mu_k & 0 \\ \mp j\mu_k & \mu_m & 0 \\ 0 & 0 & 1 \end{bmatrix}, \quad \text{where} \quad \mu_m = 1 + \omega_m(\omega_0 + i\alpha\omega) / [(\omega_0 + i\alpha\omega)^2 - \omega^2],$$

$\mu_k = \omega_m \omega / [(\omega_0 + i\alpha\omega)^2 - \omega^2]$, $\omega_m = \mu_0 \gamma M_s$, $\omega_0 = \gamma \mu_0 H_0$, and $\mu_0 H_0$ is the external magnetic field (about 0.115 Tesla) along z direction, $\gamma = 1.759 \times 10^{11}$ C/kg is the gyromagnetic ratio, $\alpha = 0.0088$ is the damping coefficient, and ω is the operating frequency³⁹.”

One new reference about the formulations of the YIG material is added on Page 13 in the revised manuscript:

“39. D. M. Pozar, *Microwave Engineering* (Wiley, 2012).”

In addition, the YIG materials selected in this paper are consistent with the previous works [*Phys. Rev. Lett.* **106**, 093903 (2011) and *Nat. Commun.* **9**, 2462 (2018)]. To make this point clear, we have provided the detail of YIG materials in the method on Page 9, starting from line 24, in the revised manuscript, which includes all information used in the simulations and experiments:

“In the experiment, we adopt commercially available yttrium iron garnet (YIG) ferrites as gyromagnetic materials to break the time-reversal symmetry. The radius and height of YIG ferrites are 1.3 mm and 2 mm, respectively. The YIG ferrites have a saturation magnetization $M_s = 1780$ Gauss. Its permittivity and permeability are about $\epsilon_r = 14.3 + 0.0002$ and $\mu_r \approx 1$, which are essentially constant at microwave frequencies. The permanent magnets (Sm2Co17) are electroplated by nickel with a thickness of 0.002 mm. One pair of magnets provide an overall uniform external magnetic field of about 0.115 Tesla to magnetize the gyromagnetic rods.”

For the YIG material resonance of the parameters selected, we have added following discussion on Page 8, starting from line 9, in the revised manuscript:

“For 0.115 T external magnetic field, the dispersions of the permeability tensor elements μ_m and μ_k are shown in Fig. S1a and Fig. S1b, respectively. It can be found that the resonance frequency of the permeability is far from the Weyl frequency, so its dispersion is rather weak

within the frequency range of interest.”

and Fig. S1 on Page 2 in the revised supplementary materials:

Fig. S1 | Permeability tensor of the gyromagnetic material. a, b Frequency-dependent elements μ_m and μ_k of the permeability tensor of the gyromagnetic material for $\mu_0 H_0 = 0.115$ T.

Referee #3 -- Comment 4:

Is the dispersion of the YIG material taken into account in the numerical simulation of the band structure? If yes how is this incorporated in the Cosmol routine?

Response from Authors:

Yes, we have considered the dispersion of the YIG material in the numerical simulations of the band structure. We incorporated the dispersive of the YIG material in COMSOL simulations by the weak form equations. This method transforms the complex band diagram problem into a simple eigenvalue problem by solving the eigenvalue with respect to wave vector k and frequency [Opt. Express **15**, 9681 (2007); Opt. Express **19**, 19027 (2011); Light Sci. Appl. **11**, 276 (2022)]. Here, we take simple two-dimensional magnetic photonic crystal [Phys. Rev. B **80**, 033105 (2009)] as an example. Firstly, an equivalent field equation is obtained by replacing the field function $E(\mathbf{r})$ with Bloch-wave function $E(\mathbf{r}) = u(\mathbf{r}) \exp[i(\omega t - \mathbf{k} \cdot \mathbf{r})]$ which makes the wavevector k become a priori unknown quantity similar to the frequency ω . Then we write the weak form of the field equation by introducing a test function, which makes sure that the electromagnetic field equation can be solved in COMSOL software. In the next step, the real structure of the unit cell (as shown in Fig. R5a) is constructed in the software, periodic boundary conditions are applied for all of the boundaries, and the eigenvalues of k are solved at fixed frequencies. The eigenvalues calculated by the weak form equations are entirely consistent with [Phys. Rev. B **80**, 033105 (2009)], as shown in Fig. R7.

Fig. R7 | Method of numerical simulation for the dispersive magnetic material. **a, b** Unit cell and simulated band structure of two-dimensional magnetic photonic crystal in [*Phys. Rev. B* **80**, 033105 (2009)]. **c** Simulated band structure by weak form equations, which are entirely consistent with that in [*Phys. Rev. B* **80**, 033105 (2009)].

Referee #3 -- Comment 5:

Could the Authors provide the raw data of the E-field distribution before the FFTs?

Response from Authors:

For both antichiral surface states on the A-type and B-type surfaces, we have plotted the measured electric field amplitude and phase distributions before the FFTs at 7.85GHz, 8.02GHz and 8.22GHz, respectively, as shown in Fig. R8, in which we can see that the antichiral surface states propagate rightward unidirectionally.

We have highlighted this point on Page 6, starting from line 5, in the revised manuscript with the sentence:

“(see Supplementary Materials for the measured electric amplitude and phase distributions of the antichiral surface states),”

Besides, Fig. R8 has been adopted as Fig. S11 on Page 16 in the revised supplementary materials.

Fig. R8 | Measured E_z field distributions of the antichiral surface states. **a-f** Measured amplitude and phase distributions of E_z when the surface states are excited by a point source (microwave dipole antenna) at the center of the A-type surface at 7.85 GHz, 8.02 GHz and 8.22 GHz, respectively. **g-l** Measured amplitude and phase distributions of E_z when the surface states are excited by a point source

(microwave dipole antenna) at the center of the B-type surface at 7.85 GHz, 8.02 GHz and 8.22 GHz, respectively.

REVIEWERS' COMMENTS

Reviewer #2 (Remarks to the Author):

In the revised paper, the authors have taken into account the comments and suggestions and improved satisfactorily the quality of the paper. Now I believe that the paper can now be accepted for publication in NC.

Reviewer #3 (Remarks to the Author):

The authors successfully address all my comments. Therefore, I recommend its publication on Nature Communications.

Reviewer #4 (Remarks to the Author):

The Authors have addressed all the comments well and the paper may now be accepted for publication

Response Letter to Referees

We are grateful for the constructive comments on our manuscript entitled “Topological antichiral surface states in a magnetic Weyl photonic crystal” (NCOMMS-22-39104A) from all three referees. The text below quotes referee comments in italics, followed by our response.

COMMENTS FROM REFEREE #1:

In the revised paper, the authors have taken into account the comments and suggestions and improved satisfactorily the quality of the paper. Now I believe that the paper can now be accepted for publication in NC.

Response from Authors:

We thank the referee for recommending the publication of our paper on Nature Communications.

COMMENTS FROM REFEREE #2:

The authors successfully address all my comments. Therefore, I recommend its publication on Nature Communications.

Response from Authors:

We thank the referee for recommending the publication of our paper on Nature Communications.

COMMENTS FROM REFEREE #3:

The Authors have addressed all the comments well and the paper may now be accepted for publication.

Response from Authors:

We thank the referee for recommending the publication of our paper on Nature Communications.